# Therapeutic Effects of *Propionibacterium acnes* and Lipopolysaccharide from *Escherichia coli* in Cats with Feline Panleukopenia

**DOI:** 10.3390/vetsci11060253

**Published:** 2024-06-04

**Authors:** Rattanakhon Chanachaivirada, Phongsakorn Chuammitri, Kannika Na Lampa, Worapat Prachasilchai, Chollada Sodarat

**Affiliations:** 1Department of Companion Animal and Wildlife Clinic, Faculty of Veterinary Medicine, Chiang Mai University, Chiang Mai 50100, Thailand; vet551410048@gmail.com (R.C.); kannika.nalampang@cmu.ac.th (K.N.L.); worapat.p@cmu.ac.th (W.P.); 2Department of Veterinary Bioscience and Veterinary Public Health, Faculty of Veterinary Medicine, Chiang Mai University, Chiang Mai 50100, Thailand; phongsakorn.c@cmu.ac.th

**Keywords:** feline panleukopenia, immunomodulator, inactivated *Propionibacterium acnes*, lipopolysaccharide from *Escherichia coli*, IgA

## Abstract

**Simple Summary:**

Feline panleukopenia (FPV) is a significant infectious disease in Thailand and, currently, there are no antiviral or immunomodulatory drugs available for effective treatment. In our study, we investigated the potential of immunomodulatory drugs, namely, *Propionibacterium acnes* and lipopolysaccharide from *Escherichia coli*, to enhance treatment outcomes for this disease. The findings revealed that the drug was capable of increasing white blood cell counts from day 3 to day 6 in the treatment group, although there was no difference in the white blood cell counts between the two groups. However, despite this immune response, the drug did not yield a significant reduction in the mortality rate. The immunoglobulin A level in serum and feces also showed no difference between the two groups.

**Abstract:**

The objective of this study was to investigate the therapeutic effects of inactivated *Propionibacterium acnes* and lipopolysaccharide derived from *Escherichia coli* cells in cats affected by feline panleukopenia virus (FPV). A retrospective study of 80 FPV-positive cats was divided into two groups: a treatment group receiving inactivated *Propionibacterium acnes* and lipopolysaccharide derived from *Escherichia coli* cells along with supportive treatment and a no-treatment group receiving only supportive treatment. There was no significant difference in the total white blood cell counts between the two groups. However, the total white blood cell counts of both groups were low on day 0 and increased significantly on days 3 and 6 of treatment. Additionally, the white blood cell counts in the treatment group significantly increased during days 3 to 6 compared with those of the no-treatment group (*p* < 0.01). The mortality rate was not significantly different between the two groups. In a prospective study, the serum and fecal immunoglobulin A (IgA) levels were measured in both groups. There were no significant differences in IgA levels between the two groups in either the serum or feces.

## 1. Introduction

Feline parvovirus (FPV) is a small, non-enveloped single-stranded DNA virus that belongs to the Parvoviridae family, specifically the parvovirus genus and feline parvovirus subgroup. Transmission of the virus from an infected cat to a susceptible cat occurs easily through the fecal–oral route [1]. Following FPV infection, viral replication takes place in the oropharyngeal epithelium within 18 to 24 h post-infection, followed by dissemination during the viremic phase within 2 to 7 days post-infection. The viremic phase ensues when the virus spreads to lymphatic organs and bone marrow, leading to cellular depletion and functional immunosuppression [2]. Complications such as circulatory shock, septicemia, and disseminated intravascular coagulation frequently lead to fatality [3]. Co-infections with enteric bacteria or enteric viruses, such as the coronavirus, can occur when the integrity of the intestinal barrier is compromised. This not only worsens the symptoms of the disease but also increases the mortality rate [4]. It is important to note that the virus, which is known to be resistant under environmental conditions, can persist in the environment for up to one year [5,6].

Currently, there are no effective drugs for the treatment of parvovirus [6]. Recombinant feline interferon omega (IFN-ω) has also shown inhibitory effects on the replication of feline parvovirus in cats in vivo [7]. Recently, horse anti-FPV IgG has been employed in certain European countries as a preventive measure against feline parvovirus infection [1]. In cases where exposure to the virus has occurred but clinical signs have not yet manifested, the administration of antiserum or high-titer parvoviral antiserum from vaccinated or recovered cats can be advantageous [8].

The use of microbial products containing killed *Propionibacterium acnes* and lipopolysaccharide from *Escherichia coli* cells can act as immunomodulators [9]. These are products that are currently under research by Laboratories Calier, 5-A, Spain. *Propionibacterium acnes*, when used as a heat- or phenol-killed suspension, acts as an immunomodulatory agent that promotes macrophage activity, resulting in the production of IL-1, interferons, and TNF [10,11]. Anecdotal reports suggest that *Propionibacterium acnes* may be beneficial in FeLV-infected cats when administered intravenously at 0.5 mL per cat twice weekly for two weeks, followed by once weekly thereafter. However, the lack of controlled studies necessitates caution when using this approach [12].

In pigs, the administration of lipopolysaccharide (LPS) and *Propionibacterium granulosum* along with a vaccine resulted in higher primary humoral responses than those in pigs receiving the vaccine alone [13]. Another study demonstrated that the combination of lipopolysaccharide (LPS) and *Propionibacterium granulosum* significantly enhanced the cytotoxicity of natural killer cells and lymphocyte proliferation in response to mitogen stimulation (*p* < 0.05) in pigs, indicating positive immunoregulation of the porcine innate immune system, non-specific activation of lymphocytes, and antibody production [14].

IgA is one of the secretory antibodies produced in large quantities by plasma cells on mucosal surfaces [15]. Secretory IgA predominates in milk, colostrum, tears, saliva, and mucosal surfaces, including those of the respiratory and gastrointestinal tract [16]. It is an important component of the immune system that plays a critical role in gastrointestinal diseases.

In the present study, we hypothesized that using the mixture of inactivated *Propionibacterium acnes* and lipopolysaccharide from *Escherichia coli* would be effective for the treatment of FPV-infected cats. To prove the hypothesis, we investigated the white blood cell counts and the IgA in naturally FPV-infected cats before and after treatment. In addition, the mortality rate was also used to evaluate the efficacy of inactivated *Propionibacterium acnes* and lipopolysaccharide from *Escherichia coli.*

## 2. Materials and Methods

### 2.1. Reagents and Product

In this study, an immunomodulatory drug, consisting of 0.25 mg inactivated *Propionibacterium acnes* and 0.02 mg of lipopolysaccharide from *Escherichia coli*, both in 1 mL solution (Laboratories Calier, Barcelona, Spain), was used.

### 2.2. Retrospective Study

#### Study Period and Data Collection

The data on cats were collected in 2022 at the Small Animal Teaching Hospital, Chiang Mai University. We included FPV-positive cats diagnosed using a rapid CPV/CCV Ag test kit (sensitivity of 100% and specificity of 98.8%, CPV/CCV Ag Test Kit, BIONOTE, Gyeonggi-do, Korea) in this study population. For the total white blood cell count analysis, the FPV-positive cats were excluded if the cats died before the third day of treatment or the cats were transferred to another hospital. The FPV-positive cats were divided into two groups: a no-treatment group and a group treated intravenously with an immunomodulator made of a 1 mL solution of inactivated *Propionibacterium acnes* and 0.02 mg of lipopolysaccharide from *Escherichia coli* for 3 consecutive days (days 0, 1, and 2). The total white blood cell counts of FPV-positive cats on days 0, 3, and 6 were obtained. The number of dead FPV-positive cats was recorded.

All FPV-positive cats were treated with fluid therapy, antibiotics (specifically amoxicillin and clavulanic acid), antiemetics (ondansetron or maropitant), antacids (such as cimetidine or omeprazole), and vitamin supplements (specifically multivitamin B).

### 2.3. Prospective Study

#### 2.3.1. Study Period and Sample Collection

Between February 2022 and April 2022, an experiment was conducted at the Small Animal Hospital of Chiang Mai University. The objective of the study was to recruit cats exhibiting clinical signs of illness associated with feline parvovirus infection, including diarrhea, vomiting, anorexia, fever, and dehydration. To be eligible for inclusion, cats had to test positive for FPV using the rapid CPV/CCV Ag test kit (sensitivity of 100% and specificity of 98.8%, CPV/CCV Ag Test Kit, BIONOTE, Korea) and all FPV-positive cats had no prior treatment. Cats that were transferred to another hospital, those that unexpectedly passed away before day 3 of treatment, and cats weighing less than 1 kg were excluded. Owners were required to sign an owner’s consent form to participate. The animal study designs and experiments used in this research were approved by the Animal Care and Use Committee (FVM-ACUC) under reference number S7/2563.

Eighteen cats were recruited and randomly assigned to two groups. The control group (*n* = 10) received only supportive therapy. The treatment group (*n* = 8) received 1 mL of immunomodulator, 1 mL of inactivated *Propionibacterium acnes*, and 0.02 mg of lipopolysaccharide from *Escherichia coli* for three consecutive days (days 0, 1, and 2) through intravenous injection, as well as supportive therapy. A total of 1 mL of whole blood from the FPV-positive cats was collected in a serum tube on day 0, day 3, and day 6. Fecal samples were obtained by rectal swab and kept in a collecting tube on day 0, day 3, and day 6.

All of them received additional medications as necessary. In this study, every cat was treated with fluid therapy, antibiotics (specifically amoxicillin and clavulanic acid), antiemetics (ondansetron or maropitant), antacids (such as cimetidine or omeprazole), and vitamin supplements (specifically multivitamin B).

#### 2.3.2. Detection of Antibodies from Serum and Feces

Serum and fecal samples were collected to measure IgA with an immuno-dot blot assay using goat anti-pig polyclonal IgA-Biotin (Bio-Rad, Hercules, CA, USA). Both serum and fecal samples were diluted with PBS (1:1) and incubated at 95 °C for 5 min. Three microliters (μL) of diluted samples were spotted onto a nitrocellulose (NC) membrane with a pore size of 0.2 μm (Bio-Rad) and air-dried. Then, the membrane was blocked with a blocking buffer (5% BSA in TSBT) for 30 min at room temperature (RT). After that, the membrane was incubated with polyclonal IgA antibody (1:3000 dilution) in blocking buffer at RT for 2 h and then incubated with undiluted Streptavidin-HRP (Cat # ab64269, Abcam, Cambridge, UK) for 10 min at RT. Finally, the membrane was stained with DAB substrate, rinsed off with distilled water, and dried at RT. Another NC membrane was prepared to determine the total protein in each serum. This NC was stained with Ponceau S (Sigma-Aldrich, St. Louis, MO, USA) staining for 10 min at RT. The Image Studio Lite TM program (LI-COR, Lincoln, NE, USA) was used to quantify the protein densitometry, which was IgA/Ponceau S. Each typical dot blot was converted into grayscale before protein densitometry was quantified [17].

### 2.4. Statistical Analysis

#### 2.4.1. Statistical Analysis of the Retrospective Study

The Shapiro–Wilk test was utilized to assess the normality of the distribution of the WBCs after applying a logarithmic transformation to the data. The mean values of white blood cells on each paired day within each group were compared using a paired *t*-test. An independent *t*-test was employed to compare the mean values of white blood cells between the treatment and control groups. Generalized estimated equations were used to examine the correlations of the total WBCs from days 0, 3, and 6 of treatment within each group. Survival curve analysis was conducted using the Kaplan–Meier method. For statistical analysis, R packages version 4.4.0, SPSS version 29.0.2.0, and GraphPad Prism version 9 were utilized. Results with a significance level of *p* < 0.05 were considered statistically significant.

#### 2.4.2. Statistical Analysis of the Prospective Study

After applying a logarithmic transformation to the data, the Shapiro–Wilk test was utilized to assess the normality of the distribution of IgA in the serum and fecal samples. An independent *t*-test was employed to determine the mean values of serum and fecal IgA between the treatment and control groups. To examine the correlations of IgA in blood and fecal samples across day 0, day 3, and day 6 of treatment within each group, a generalized estimated equation was utilized. Statistical analyses were conducted using R packages version 4.4.0, SPSS version 29.0.2.0. A result with a *p*-value lower than 0.05 was considered statistically significant.

## 3. Results

### 3.1. Retrospective Study

#### 3.1.1. Descriptive Analysis

From 1 January to 31 December 2022, 153 feline cases at the Small Animal Hospital, Chiang Mai University, were diagnosed with feline parvovirus infection. All FPV-positive cats received additional medications as necessary. In this study, every cat was treated with fluid therapy, antibiotics (specifically amoxicillin and clavulanic acid), antiemetics (ondansetron or maropitant), antacids (such as cimetidine or omeprazole), and vitamin supplements (specifically multivitamin B).

The FPV-positive cats consisted of 87 male cats and 66 female cats, with an average age of 11.80 months and an average body weight of 2.47 kg. The breed distribution revealed that there were 125 Domestic Shorthair cats, 15 mixed-breed cats, 7 Scottish Fold cats, 4 Persian cats, 1 American Shorthair cat, and 1 Maine Coon cat. Forty-seven FPV-positive cats died before completing three days of treatment; thus, they were excluded from the WBC analysis. Twenty-six FPV-positive cats were referred to another hospital; consequently, they were excluded from the survival and WBC analyses.

In the retrospective study, 80 FPV-positive cats met the inclusion criteria for the WBC analysis. The FPV-positive cats were categorized into two groups: the no-treatment group (45 cats on day 0, 45 cats on day 3, and 27 cats on day 6) and the group undergoing treatment with killed *Propionibacterium acnes* and lipopolysaccharide from *Escherichia coli* cells (35 cats on day 0, 35 cats on day 3, and 24 cats on day 6). The data on WBCs were collected on days 0, 3, and 6. The WBCs in FPV-positive cats were low on day 0, but those of almost all of the cats increased on days 3 and 6 (Figure 1).

#### 3.1.2. Comparison of the Mean Total White Blood Cell Counts between and within Groups

The WBCs in the treatment group were significantly different (*p* < 0.01) between day 0 and day 3, day 0 and day 6, and day 3 and day 6, while the WBCs in the no-treatment group were significantly different only between day 0 and day 3 (*p* < 0.05) and between day 0 and day 6 (*p* < 0.01) (Table 1). However, there was no significant difference between day 3 and day 6 in the no-treatment group. Nevertheless, the WBC counts in both groups on day 0, day 3, and day 6 were not significantly different (Figure 2).

#### 3.1.3. Investigation of Repeated Measurements

According to the comparison using the generalized estimation equation within the groups, leukocytopenia was found in both groups on the day of admission (day 0). In the no-treatment group, there was a statistically significant increase (*p* < 0.01) in the Log10-transformed total white blood cell counts between days 0 and 3 (0.69 units) and days 0 and day 6 (0.81 units). Similarly, in the treatment group, there was a significant increase (*p* < 0.01) in the Log10-transformed total white blood cell counts between days 0 and 3 (1.28 units) and days 0 and 6 (1.78 units) (Table 2).

#### 3.1.4. Survival Curve Analysis

A Kaplan–Meier survival curve was calculated based on the data from 127 FPV-positive cats (Figure 1). The mortality rate of the 127 FPV-positive cats was 40.94% (52/127), and it was 38.00% (19/50) in the treatment group and 42.86% (33/77) in the no-treatment group. However, the survival curve indicated that there were no significant differences between the treatment and no-treatment groups (Figure 3). There was a high mortality rate in both groups during the first five days after diagnosis (Figure 3).

The results of a paired *t*-test were analyzed to assess the mean total white blood cell counts at different time points within the same group of cats. The WBCs in the treatment group were significantly different (*p* < 0.01) between day 0 and day 3, day 0 and day 6, and day 3 and day 6, while the WBCs in the no-treatment group were significantly different only between day 0 and day 3 (*p* < 0.05) and between day 0 and day 6 (*p* < 0.01).

### 3.2. Prospective Study

#### 3.2.1. Descriptive Analysis

Eighteen FPV-positive cats met the inclusion criteria. They were divided into two groups: the control group (10 cats) and the group undergoing treatment with killed *Propionibacterium acnes* and lipopolysaccharide from *Escherichia coli* cells (8 cats). In the control group, the average age of the cats was 8.8 months, ranging from 3 to 24 months, with a distribution of seven males and three females, while the average age of the cats in the treatment group was 10.5 months, ranging from 3 to 24 months, with a distribution of three males and five females. Two cats from the control group and one cat from the treatment group died during the study.

#### 3.2.2. Detection of Antibodies in the Serum and Feces

Samples of serum and feces were repeatedly collected to assess the concentration of IgA using dot blotting. There were no significant differences in the levels of serum IgA between both groups (Table 3). However, a notable finding was that the serum IgA in the treatment group significantly decreased on day 3 compared with that on day 0 (*p* < 0.01) (Table 4). Similarly, no significant differences were observed in the levels of fecal IgA between both groups (Table 5). Nonetheless, the levels of fecal IgA on day 3 and day 6 showed no significant differences from day 0 in both groups (Table 6).

## 4. Discussion

There was no significant impact of gender on the FPV-positive cats [18]. However, male cats tended to outnumber female cats. The majority of cats with FPV were found to roam freely and live in households with multiple cats. Only one cat was reported to live exclusively indoors, and the time of infection for this particular cat remained unknown. Contrarily, a prior study found no correlation between living conditions and the prognosis factor [19].

PCR is considered the gold standard for parvovirus testing [20]. However, PCR testing is time-consuming, expensive, and technically challenging [20]. Previous research has indicated that test kits designed for canine parvovirus can also detect feline parvovirus in sick cats [20,21]. Hence, we utilized CPV rapid antigen test kits in our study for diagnosis. Every cat in our study exhibited clinical symptoms and had undergone screening using a rapid CPV antigen test kit.

The survival rate of FPV-positive cats in our study was 59.06%, which was slightly different from that in research conducted in Germany in 2010 (51.10%) and in Indonesia in 2017 (55%) [19,22]. The survival rate in the treatment group, 62% (31/50), was higher than the survival rate in the no-treatment group, 57.14% (44/77), but this was not statistically significant. The severity of leukopenia is correlated with the manifestation of severe symptoms and an increased mortality rate in FPV-positive cats [19]. Non-surviving FPV-positive cats were found to have lower leukocyte counts on days 3, 4, and 7 of hospitalization in comparison with surviving FPV-positive cats [23]. In this study, all FPV-positive cats initially showed leukopenia and there was a significant increase in WBCs in-both the treatment and no-treatment groups on days 3 and 6 of admission (*p* < 0.01). However, only the WBCs in the treatment group increased significantly from day 3 to day 6. This finding suggests that cats who receive immunomodulators may exhibit a robust immune response when infected with FPV. It is worth noting that *Propionibacterium acnes* has been found to possess immunomodulatory properties, resulting in the up-regulation of type-1 cytokine genes, interferon synthesis, the simulation of macrophage activity, natural killer cell cytotoxicity, and IL-1 production [24]. Moreover, the total leukocyte count, granulocyte count, and phagocytic activity in animals were found to significantly increase in response to LPS derived from normal isolates of *Escherichia coli* [25].

Plasma cells in the intestinal lamina propria produce IgA, which is crucial for combating gastrointestinal pathogens. Fecal IgA concentration is significantly associated with IgA levels in duodenal explant cultures [26]. When feline parvovirus destroys intestinal villi, bacteria can enter the bloodstream [26,27]. This study investigated the impacts of inactivated *Propionibacterium acnes* and *Escherichia coli* lipopolysaccharide on IgA concentrations in serum and feces. Interestingly, the serum IgA levels in the treatment group were significantly lower (*p* < 0.01) on day 3 than on day 0. Generally, the synthesis of serum IgA occurs in the bone marrow. Plasma cells in the spleen, lymph nodes, and peripheral blood also produce IgA [28]. Reductions in serum IgA levels may be due to the spread of FPV to lymphoid tissue and bone marrow. The fecal IgA levels in both groups were not significantly different because FPV destroyed the lamina propria.

FPV-positive cats have a poor prognosis, with a median life expectancy of only 3 days [24]. Therefore, our study was designed to observe the cats within a 6-day treatment period. FPV-positive cats died due to sepsis, disseminated intravascular coagulopathy, secondary bacterial infection, and dehydration [19]. Effective treatment involves early identification of the infection and aggressive therapy, including antiviral or immunomodulatory medication, and intensive care should be applied to increase the survival rate in FPV-positive cats [29]. In our study, the mortality rate decreased dramatically after the fifth day of hospitalization. However, our studies did not find a significant difference in the mortality rate between the two groups with the survival curve. Currently, there is no effective antiviral drug or immunomodulatory therapy that reduces the mortality rate of FPV.

## 5. Conclusions

The intravenous administration of the solution consisting of 0.25 mg inactivated *Propionibacterium acnes* and 0.02 mg of lipopolysaccharide from *Escherichia coli*, 1 mL/cat for 3 consecutive days, did not significantly reduce the mortality rates in FPV-positive cats. The white blood cell counts showed no significant difference between the treatment and control groups. Additionally, no significant difference was observed in the effects of immunomodulators on IgA levels in serum and feces between the two groups. The results of the present study suggest that, in FPV-positive cats, the mixture of inactivated *Propionibacterium acnes* and lipopolysaccharide from *Escherichia coli* does not have therapeutic properties.

## Figures and Tables

**Figure 1 vetsci-11-00253-f001:**
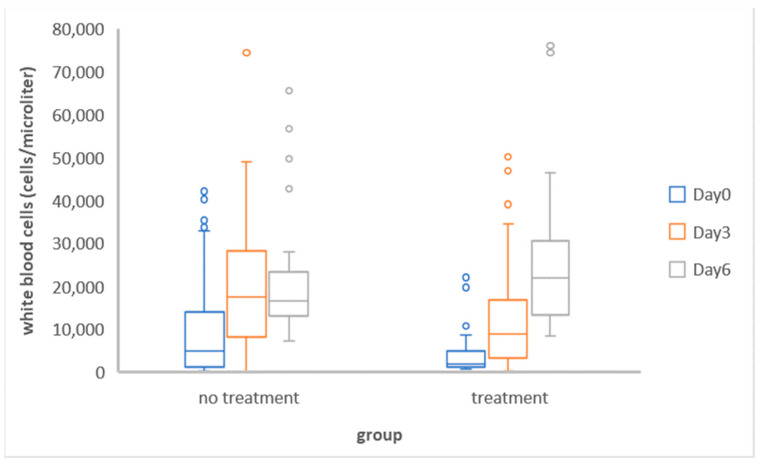
White blood cell counts of groups of FPV-positive cats with no treatment and with treatment on day 0, day 3, and day 6.

**Figure 2 vetsci-11-00253-f002:**
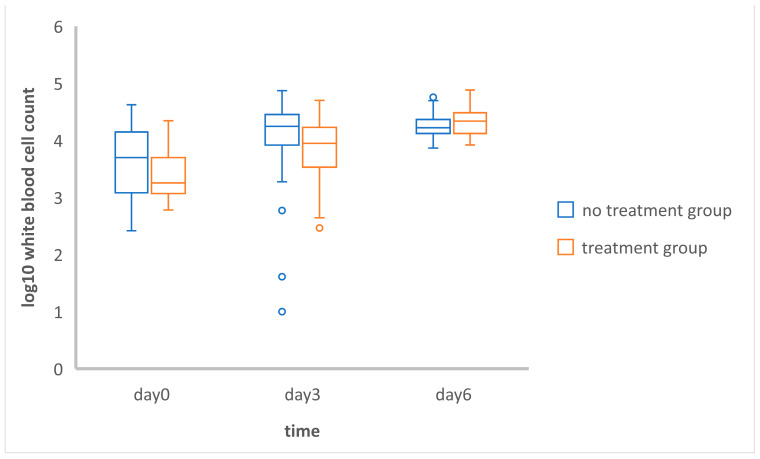
Comparison of Log10 values of the white blood cell counts in the no treatment and treatment group on days 0, 3, and 6. The independent *t*-test conducted in the analysis indicated that there were no statistically significant differences observed in the mean of the Log10-transformed white blood cell counts on days 0, 3, and 6 between the no-treatment and treatment groups.

**Figure 3 vetsci-11-00253-f003:**
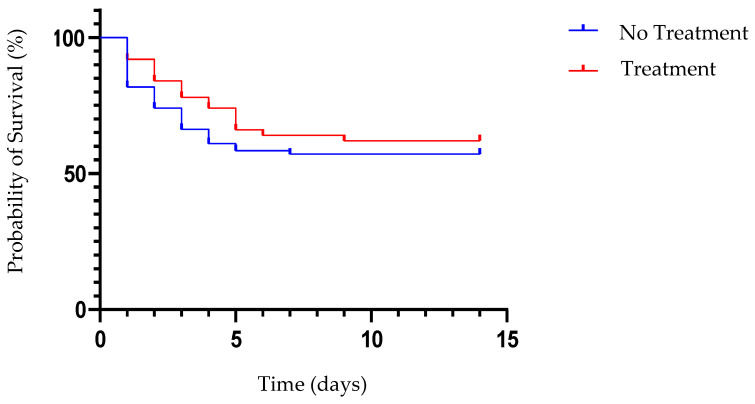
The Kaplan–Meier survival curve of FPV-positive cats. A comparison between the survival probabilities of the control and treatment groups is shown. Within five days, the likelihood of survival dropped to about 50% in both groups.

**Table 1 vetsci-11-00253-t001:** Comparison of the white blood cell counts at different time points within the same group.

Group	Day	Mean of Log10 WBC	SD	Mean Difference	SD of Difference	t	*p*
No Treatment	D-0	3.64	0.64	−0.40	0.99	−2.70	<0.05
	D-3	4.04	0.72				
	D-0	3.58	0.59	−0.69	0.65	−5.56	<0.01
	D-6	4.27	0.25				
	D-3	4.27	0.29	0.00	0.34	0.02	0.98
	D-6	4.27	0.25				
Treatment	D-0	3.41	0.45	−0.40	0.68	−3.50	<0.01
	D-3	3.81	0.60				
	D-0	3.31	0.39	−1.04	0.46	−11.14	<0.01
	D-6	4.35	0.26				
	D-3	3.71	0.66	−0.64	0.71	−4.39	<0.01
	D-6	4.35	0.26				

**Table 2 vetsci-11-00253-t002:** Results of the generalized estimation equation for total white blood cell counts within groups.

WBC Coefficients	Compared Days	Estimate	SE	Wald	Pr (>|W|)
No treatment (*n* = 45)	0–3	0.69	0.21	10.65	<0.01
	0–6	0.81	0.22	13.86	<0.01
Treatment (*n* = 35)	0–3	1.28	0.34	13.96	<0.01
	0–6	1.78	0.25	49.26	<0.01

In the no-treatment group, there was a statistically significant increase (*p* < 0.01) in the Log10-transformed total white blood cell counts between days 0 and 3 (0.69 units) and days 0 and day 6 (0.81 units). Similarly, in the treatment group, there was a significant increase (*p* < 0.01) in the Log10-transformed total white blood cell counts between days 0 and 3 (1.28 units) and days 0 and 6 (1.78 units).

**Table 3 vetsci-11-00253-t003:** Comparison of serum IgA levels between the control and treatment groups.

Day	Group	Mean of Log10 Serum IgA	SD	t	*p*
0	Control (*n* = 10)	−1.10	1.09	−0.60	0.56
	Treatment (*n* = 8)	−0.81	0.66		
3	Control (*n* = 10)	−1.27	0.87	0.18	0.86
	Treatment (*n* = 7)	−1.37	0.11		
6	Control (*n* = 8)	−0.72	0.64	0.18	0.86
	Treatment (*n* = 7)	−0.79	0.61		

An independent *t*-test was conducted to compare serum IgA levels between the control and treatment groups. The analysis revealed no significant differences between the two groups on days 0, 3, and 6.

**Table 4 vetsci-11-00253-t004:** Comparison of the serum IgA levels between days 0 and 3 and days 0 and 6 in the control and treatment groups.

Serum Coefficients	Comparison Days	Estimate	SE	Wald	Pr (>|W|)
Control	0–3	−0.52	0.83	0.40	0.53
	0–6	0.51	0.50	1.06	0.30
Treatment	0–3	−2.64	0.58	20.71	<0.01
	0–6	−0.604	0.56	1.14	0.28

The generalized estimation equation was employed. In the control group, there were no significant differences observed in serum IgA levels on day 3 and day 6 compared to day 0. Conversely, in the treatment group, serum IgA levels on day 3 were significantly lower (*p* < 0.01) than those on day 0 and showed no significant difference compared to day 6 when compared to day 0.

**Table 5 vetsci-11-00253-t005:** Comparison of the fecal IgA levels between the control and treatment groups.

Day	Group	Mean of Log10 Feces IgA	sd	t	*p*
0	Control (*n* = 10)	−0.55	0.57	0.06	0.95
	Treatment (*n* = 8)	−0.57	0.55		
3	Control (*n* = 10)	−0.62	0.50	−0.09	0.93
	Treatment (*n* = 7)	−0.60	0.29		
6	Control (*n* = 8)	−0.51	0.69	−0.36	0.74
	Treatment (*n* = 7)	−0.39	0.32		

An independent *t*-test was conducted to compare fecal IgA levels between the control and treatment groups. The analysis revealed no significant differences between the two groups on days 0, 3, and 6.

**Table 6 vetsci-11-00253-t006:** Comparison of the fecal IgA levels between days 0 and 3 and days 0 and 6 in the control and treatment groups.

Feces Coefficient	Comparison Days	Estimate	SE	Wald	Pr (>|W|)
Control (*n* = 10)	0–3	−0.01	0.55	0.00	0.98
	0–6	0.33	0.58	0.33	0.57
Treatment (*n* = 8)	0–3	−0.01	0.65	0.00	0.98
	0–6	0.70	0.63	1.24	0.26

The generalized estimation equation was employed. There were no significant differences observed in fecal IgA levels in the two groups on day 3 and day 6 compared to day 0.

## Data Availability

Data related to this study are available within the article.

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
