# Peer review of "Therapeutic Effects of Propionibacterium acnes and Lipopolysaccharide from Escherichia coli in Cats with Feline Panleukopenia"

_vetsci, 2024, doi:10.3390/vetsci11060253_

Round 1

Reviewer 1 Report

Comments and Suggestions for Authors

The manuscript regards the Therapeutic Effect of Propionibacterium acnes and Lipopolysaccharide from Escherichia coli in Cats with Feline Panleukopenia. Overall the manuscript seems very interesting. The information obtained is important to the field and can help improve treatment of the animals with FILV and FELV.

The results are very long and maybe figures and not tables would be more interesting.

Some sections are in red, please correct them.

sera or serum?

The references need corrections according to the guidelines of the journal.

Author Response

Dear sir,

Thank you for taking the time to assess my manuscript.  Your suggestions and comments are highly appreciated. We have revised the manuscript already.  I changed some result tables to figures.  The red section was already corrected. The serum was used in our manuscript. The reference was corrected according to the guidelines of the journal.  In addition, all spelling and grammatical errors have been corrected using English language editing by MDPI.

We look forward to hearing from you and to respond to any further questions and comments you may have.

Sincerely,

Chollada Sodarat

Reviewer 2 Report

Comments and Suggestions for Authors

Overall Comments:

     In this study, the authors aimed to develop a treatment plan to enhance outcomes and reduce mortality rates in infected cats in the future. They assessed the therapeutic potential of inactivated Propionibacterium acnes and Lipopolysaccharide from Escherichia coli cells in naturally infected cats with FPV. The results indicated that the treatment could boost the immune response in sick cats, demonstrated by an increase in their white blood cell count compared to the untreated group. However, despite this immune response, the treatment did not lead to a significant decrease in mortality rates. Therefore, they recommend further investigation to explore the prophylactic potential of Propionibacterium acnes and Lipopolysaccharide from Escherichia coli in preventing feline panleukopenia.

     However, the manuscript lacks novelty overall, has a disorganized structure, significant flaws in experimental design, and poor writing quality. The results presented do not adequately support the conclusions drawn by the authors. Additionally, the manuscript lacks proper certification, with issues not only in language and writing but also in formatting. Specific issues, not limited to the following, include:

1)     The logical flow of the manuscript is poor, with abrupt transitions between paragraphs. To improve overall readability, a more coherent structure and smoother transitions are essential. Additionally, refining the writing style for clarity and precision will contribute to conveying complex ideas more effectively.

2)     The overall structure of the manuscript is disorganized, with the Statistical Analysis section needing to be consolidated into a single paragraph rather than two separate ones.

3)     The authors transition abruptly from one aspect to another without appropriate transitions.

4)     Figure 1 lacks a y-axis, contains errors in plotting, and fails to display some content.

5)     There are an excessive number of tables in the manuscript, some of which may need to be presented as figures for better understanding by reviewers and readers.

6)     In Line 297, the authors reference a 25th citation that is missing from the reference list.

7)     Format issue in Line 162.

8)     Lines 309-334 suggest that the authors may have rushed the submission of this manuscript without thorough preparation.

9)     The article could benefit from a thorough proofreading and editing to improve the clarity and coherence of the writing. This includes addressing any grammatical errors, improving sentence structure, and ensuring consistency in terminology and formatting.

Comments on the Quality of English Language

English very difficult to understand.

Author Response

Dear sir,

I would like to thank you for your valuable suggestions and comments. I have revised the manuscript already.

  • The logical flow of the manuscript is poor, with abrupt transitions between paragraphs. To improve overall readability, a more coherent structure and smoother transitions are essential. Additionally, refining the writing style for clarity and precision will contribute to conveying complex ideas more effectively.

The logical flow of the manuscript and transitions between paragraphs were edited throughout the manuscript.

  • The overall structure of the manuscript is disorganized, with the Statistical Analysis section needing to be consolidated into a single paragraph rather than two separate ones.

I already added the Statistical Analysis within one section.

  • The authors transition abruptly from one aspect to another without appropriate transitions.

I revised the manuscript to make smooth and appropriate transitions

4)     Figure 1 lacks a y-axis, contains errors in plotting, and fails to display some content.      

       Figure 1 was changed to Figure 3. A y-axis explanation was already written. I used Graphpad prism 9 to make a survival curve graph. The graph came out after I filled in all the data of survivor FPV-positive cats in the program.

5)     There are an excessive number of tables in the manuscript, some of which may need to be presented as figures for better understanding by reviewers and readers.

      I agree so Table 2 and Table 3 were changed to Figure 1 and Figure 2.

6)     In Line 297, the authors reference a 25th citation that is missing from the reference list.

      I corrected this mistake already.

7)     Format issue in Line 162.

      I have revised it already.

8)     Lines 309-334 suggest that the authors may have rushed the submission of this manuscript without thorough preparation.

      I have revised the manuscript already.

9)     The article could benefit from thorough proofreading and editing to improve the clarity and coherence of the writing. This includes addressing any grammatical errors, improving sentence structure, and ensuring consistency in terminology and formatting.

In addition, all spelling and grammatical errors have been corrected using English language editing by MDPI .

We look forward to hearing from you and to respond to any further questions and comments you may have.

Sincerely,

Chollada Sodarat

Reviewer 3 Report

Comments and Suggestions for Authors

The manuscript titled "Therapeutic Effect of Propionibacterium acnes and Lipopoly- saccharide from Escherichia coli in Cats with Feline Panleuko-penia" is an interesting study, however, authors concluded that inactivated Propionibacterium acnes and Escherichia coli lipopolysaccharide, when combined with intensive treatment care, may act as effective immunotherapy stimulants, leading to a significant increase in total white blood cell counts (p<0.01), but they do not have an effect on lowering mortality rates. After a careful review of the manuscript, this reviewer has following suggestions: 

L61: Please provide citations. 

L65: More citations needed. 

L67-70: More citations needed.

L71-74: Citations needed. 

L78-83: Citations needed. 

L95: "Data from cats diagnosed with FPV infection using a positive rapid CPV/CCV Ag test kit." This is incomplete sentence. 

Methods: 2.2.1 Study period and sample collection: How many cats were sampled? Any stats on the minimum number of cats required for the statistical analyses? 

Methods: This section needs more information on the clinical symptoms of illness in cats under study. Please mention the reason of the mortality of cats, with more information on the percent mortality etc. Were dead cats necropsied? Please provide in-depth information on the methods used along with catalogue number(s) of kits used in this study. Please describe the methods in detail.

L118: 'exhibiting clinical signs'. Please note that animal do not exhibit symptoms, they rather exhibit clinical signs of illness.

L120: Is this a standard test kit used in veterinary clinics? Please provide any information of its specificity and sensitivity.

Methods: 2.3.2 Please provide information if a previously reported protocol was followed, with citation(s). 

L162: Retrospective. Remove the point before the word.

Results: 3.1.1 Any brief information on the treatment given to the cats?

Results: Tables 1, 2, 3, and 4 are not discussed. Please discuss these results where presented.

 Again, Tables 5 to 9 are not discussed at all. Data of the tables should be briefly explained and discussed wherever the tables are presented in the results section. 

Also, please provide a brief paragraph describing the results before the start of the Discussion section. 

L256: "In the retrospective study, it was observed that there was no significant impact of gender on FPV. However, male cats tended to outnumber female cats." Such sentences must be improved for better reading. 

L292: "Cats infected with FPV have a poor prognosis, with a median life expectancy of only 3 days". There are many claims without citations, such as this one. 

Other mandatory sections of the manuscript such as author contributions, funding information, data availability statement etc are incomplete. 

To summarize, sections of this manuscript need a major revision. Authors need to provide more details of the methods used. Also, the manuscript severely lacks the citations provided. 

Comments on the Quality of English Language

English requires an extensive editing, as far as grammar is concerned. 

Author Response

Dear Sir,

We would like to thank you for a valuable suggestions and comments. We have revised the manuscript already.

L61: Please provide citations. 

We deleted “Feline parvovirus infection cases are consistently observed throughout the year in Thailand.”  from manuscript. There no published paper but as the veterinarian in the veterinary hospital in Thailand for 20 years; FPV-positive cats were found throughout the year.

L65: More citations needed. 

The other one paper was added in L57

L67-70: More citations needed.

We searched the reference about Recombinant feline interferon omega (IFN-ω) and panleukopenia virus in vivo. I found only one previous paper that we referred to in L60 There is few studies about the effects of Recombinant feline interferon omega (IFN-ω) on FPV-positive cats.  

L71-74: Citations needed. 

 The reference was added already. (L61-62)

L78-83: Citations needed. 

We deleted this sentence due to It is from the distributor ‘s leatlet.

L95: "Data from cats diagnosed with FPV infection using a positive rapid CPV/CCV Ag test kit." This is incomplete sentence. 

We have revised the manuscript already.

Methods: 2.2.1 Study period and sample collection: How many cats were sampled? Any stats on the minimum number of cats required for the statistical analyses? 

During the study period, 153 cats with clinical illness signs were diagnosed with FPV using test kit. Forty -seven FP-positive cats died before third day of admission and 26 FPV-positive cat were referred to other hospital. This was show in result; retrospective study; descriptive analysis part. We use the G*power program to calculate the minimum required number per group. Twenty-nine cats per groups were acceptable.  

Methods: This section needs more information on the clinical symptoms of illness in cats under study. Please mention the reason of the mortality of cats, with more information on the percent mortality etc. Were dead cats necropsied? Please provide in-depth information on the methods used along with catalogue number(s) of kits used in this study. Please describe the methods in detail.

The clinical symptom of illness in cats in our study was recorded in OPD but we were not obtained this data. All of illness cats showed severity of anorexia, hypersalivation, vomiting, diarrhea (watery or bloody diarrhea) and weakness although we gave an aggressive therapy such as intravenous fluid, collecting electrolyte disturbance, antiemetic drug, intravenous nutrition and vitamin support and intravenous antibiotic.  After the cat died, we did not do necropsied.   

The in-depth information on the methods used along with catalogue number(s) of kits used in this study. We collected feces from rectum of suspect cat using cotton bud then dip it in reagent. Four drop of Mixture reagent was applied kit. If it is positive for FPV, band will be shown at T bar.

L118: 'exhibiting clinical signs'. Please note that animal do not exhibit symptoms, they rather exhibit clinical signs of illness.

I have revised in manuscript. (L110)

L120: Is this a standard test kit used in veterinary clinics? Please provide any information of its specificity and sensitivity.

This test kit was used for diagnosis FPV in our hospital for several year. It is widespread used in Thailand.  The specificity and sensitivity of test was 98.8% and 100% respectively.

Methods: 2.3.2 Please provide information if a previously reported protocol was followed, with citation(s). 

The protocol was used in our laboratory. There are no previous reported protocol but It was accepted to use in publish paper ‘Evaluation of Bcl-2 as a marker for chronic kidney disease prediction in cats’.

L162: Retrospective. Remove the point before the word.

I revised manuscript already.

Results: 3.1.1 Any brief information on the treatment given to the cats?

We adds the treatment information on L169-173.

Results: Tables 1, 2, 3, and 4 are not discussed. Please discuss these results where presented.

 Again, Tables 5 to 9 are not discussed at all. Data of the tables should be briefly explained and discussed wherever the tables are presented in the results section. 

Also, please provide a brief paragraph describing the results before the start of the Discussion section. 

I revised manuscript and added the brief paragraph describing the results.

L256: "In the retrospective study, it was observed that there was no significant impact of gender on FPV. However, male cats tended to outnumber female cats." Such sentences must be improved for better reading. 

I revised manuscript already.

L292: "Cats infected with FPV have a poor prognosis, with a median life expectancy of only 3 days". There are many claims without citations, such as this one. 

I deleted this sentence and discuss on L301-311

Other mandatory sections of the manuscript such as author contributions, funding information, data availability statement etc are incomplete. 

I edited this part already on L349-369.

In addition to the above suggestions and comments, all grammatical and sentence structure errors was corrected by English language editing by MDPI.

We look forward to hearing from you and to respond to any further questions and comments you may have.

Sincerely,

Chollada Sodarat

Reviewer 4 Report

Comments and Suggestions for Authors

The manuscript is well written and structured. Anyway, some little changes have to be performed in order to go ongoing with the publication. 

  • This study explored a new potential treatment for FPV by using immunomodulation , tracking white blood cell counts and IgA levels and providing valuable data on the immune response.

  • The primary goal (reducing mortality) wasn't achieved by the treatment, so I suggest the Authors to better focus this aspect, in particular in the conclusion, not just reporting that further studies must be conducted. but for example already give possible solution to conduct future studies and where address more investigations. 
  • In particular, the discussion must be enriched reporting these consideration. For example, a bigger group of cats would provide more statistically relevant data on whether the treatment impacts mortality; monitoring the cats for a longer period (weeks or months) might reveal if the initial white blood cell increase translates to improved survival rates; considering the severity of FPV in each cat could help determine if the treatment is more effective in mild or severe cases. These points must be analysed by the Authors, reporting comparisons with past studies and giving a personal and complete overview. 
  • the introduction section is too short. Please search to enrich it also reporting more references. The firs part is quite general, focus on drug treatments and evolution of the management of the disease from a clinic and therapeutic point of view. 

Author Response

Dear sir,

We would to thank you for taking the time to assess my manuscript. Your suggestions and comments are highly appreciated. I have revised the manuscript. We wrote the new information in the introduction for the manuscript. The minimum required number of FPV-positive cats per group was calculated using the G*power program. Twenty-nine cats per group were acceptable. The randomized selected FPV-positive cats in each group may reduce the factors that influence the mortality rate.

In addition, all spelling and grammatical errors have been corrected using English language editing by MDPI .

We look forward to hearing from you and to respond to any further questions and comments you may have.

Sincerely,

Chollada Sodarat

Round 2

Reviewer 2 Report

Comments and Suggestions for Authors

Overall Comments:

    The author did not take the reviewers' comments seriously, and did not make thorough and comprehensive revisions to the issues pointed out by the referees.

    Additionally, the reviewers identified the main issues in the article and the specific locations (line no.) where these issues were found, but the author only responded briefly to most of the referees' comments without indicating the specific locations of the revisions.

    If the author does not add new results and thoroughly address the issues raised by the referees, this draft is not recommended for publication.

Comments on the Quality of English Language

The level of English writing has improved.

Author Response

Dear Sir,

Firstly, I would like to apologize that I did not revise the manuscript to cover all the topics of your comments and suggestions. Then, thank you for giving me the opportunity to submit the second revised manuscript.  Now I have revised the manuscript. The revisions of the manuscript were highlighted. We have added new information to the introduction. The cited references were edited to be relevant to the research.

In Thailand, a mixture of inactivated Propionibacterium acnes and lipopolysaccharide from Escherichia coli was used widely to treat FPV-positive cats. Our research hypothesis is that a mixture of inactivated Propionibacterium acnes and lipopolysaccharide from Escherichia coli would be effective for the treatment of FPV-infected cats. We investigated the white blood cell counts and the IgA levels in blood and feces in naturally FPV-infected cats before and after treatment. In addition, the mortality rate was also used to evaluate the efficacy of inactivated Propionibacterium acnes and lipopolysaccharide from Escherichia coli. We use the G*power program to calculate the minimum required number per group. Twenty-nine cats per group were acceptable. Due to the limits of “in-field” studies it is difficult to include FPV-positive cats in the experiment. Finally, we designed to separate the study into two parts; a prospective and a retrospective study. We used an independent t-test to compare treatment and no-treatment groups (in a retrospective study) and control and treatment groups (in a prospective study). We described the method in the materials and methods part. We wrote the new conclusions in L 362-370. However, we did not have the new results. We think our results can answer our research question.

We look forward to hearing from you.

Sincerely,

Chollada Sodarat

Reviewer 3 Report

Comments and Suggestions for Authors

Thanks for revising and considering my suggestions. Well done! 

Author Response

Dear Sir, 

I would like to thank you for your valuable feedback on my manuscript. 

sincerely

Chollada sodarat

Round 3

Reviewer 2 Report

Comments and Suggestions for Authors

After numerous revisions by the author, the quality of this article has been improved.